# Bioisosteric Replacement in the Search for Biologically Active Compounds: Design, Synthesis and Anti-Inflammatory Activity of Novel [1,2,4]triazino[2,3-c]quinazolines

**DOI:** 10.3390/ph17111437

**Published:** 2024-10-28

**Authors:** Oleksandr Grytsak, Kostiantyn Shabelnyk, Hanna Severina, Victor Ryzhenko, Oleksii Voskoboinik, Igor Belenichev, Serhii Kovalenko, Valentyn Oksenych, Oleksandr Kamyshnyi

**Affiliations:** 1Department of Pharmaceutical, Organic and Bioorganic Chemistry, Zaporizhzhia State Medical and Pharmaceutical University, 69000 Zaporizhzhia, Ukraine; 2Department of Pharmaceutical Chemistry, National University of Pharmacy, 61168 Kharkiv, Ukraine; 3Department of Medical and Pharmaceutical Informatics and Modern Technologies, Zaporizhzhia State Medical and Pharmaceutical University, 69035 Zaporizhzhia, Ukraine; 4Department of Composite Materials, Chemistry and Technologies, National University «Zaporizhzhia Polytechnic», 69063 Zaporizhzhia, Ukraine; a.yu.voskoboynik@gmail.com; 5Department of Pharmacology and Medical Formulation with Course of Normal Physiology, Zaporizhzhia State Medical and Pharmaceutical University, 69000 Zaporizhzhia, Ukraine; 6Institute of Chemistry and Geology, Oles Honchar Dnipro National University, 49000 Dnipro, Ukraine; 7Department of Clinical Science, University of Bergen, 5020 Bergen, Norway; 8Department of Clinical and Molecular Medicine, Norwegian University of Science and Technology (NTNU), 7028 Trondheim, Norway; 9Department of Biosciences and Nutrition, Karolinska Institutet, 14183 Huddinge, Sweden; 10Department of Microbiology, Virology and Immunology, I. Horbachevsky Ternopil State Medical University, 46001 Ternopil, Ukraine

**Keywords:** bioisosteric replacement, molecular docking, anti-inflammatory activity, inflammation markers, ADME analysis

## Abstract

**Background:** Designing novel biologically active compounds with anti-inflammatory properties based on condensed quinazolines is a significant area of interest in modern medicinal chemistry. In the present study, we describe the development of promising new bioactive molecules through the bioisosteric replacement of a carbon atom with a sulfur atom in anti-inflammatory agents, specifically 3-methyl-2-oxo-2*H*-[1,2,4]triazino[2,3-*c*]quinazolin-6-yl)butanoate. **Methods:** Design and synthetic studies have led to the series of previously unknown substituted 2-[((3-R-2-oxo-2*H*-[1,2,4]triazino[2,3-*c*]quinazolin-6-yl)methyl)thio]carboxylic acids and their esters. These compounds were synthesized by reacting 6-chloroalkyl-3-R-2*H*-[1,2,4]triazino[2,3-*c*]quinazolin-2-ones with sulfanylalkyl carboxylic acids and their functional derivatives. The purity and structure of the obtained compounds were confirmed using a set of physicochemical methods, including elemental analysis, HPLC-MS, and ^1^H NMR spectroscopy. Molecular modeling, predicted toxicity, drug-likeness, and pharmacokinetics data were used to select compounds for evaluation of their effects on acute aseptic inflammation (carrageenan-induced paw edema test) and on markers of the inflammatory process. **Results:** The compound 2-((1-(3-methyl-2-oxo-2*H*-[1,2,4]triazino[2,3-*c*]quinazolin-6-yl)ethyl)thio)acetic acid (compound **2e**) was identified as the most active anti-inflammatory agent (AA = 53.41%), demonstrating significant inhibition of both paw edema development and the generation of pro-inflammatory cytokines and mediators. **Conclusions:** Results from docking studies and analysis of “structure-affinity” correlations revealed that these compounds are promising candidates for further modification and detailed investigation of their anti-inflammatory activity

## 1. Introduction

Quinazoline is a privileged scaffold in medicinal chemistry and a unique pharmacophore that exhibits a wide range of biological activities and is present in numerous approved and investigational drugs [1,2,3,4,5,6,7,8,9,10]. Condensed quinazolines are of high importance as subjects of studies aimed at the creation of novel bioactive compounds [11,12,13,14,15]. The chemical nature of quinazolines and their condensed derivatives determines the tremendous variety of synthetic approaches that allow the elaboration of novel original compounds, expanding the chemical and pharmacological space [10,12,15,16,17,18,19,20]. Thus, compounds with high anticancer [2,5,10,21,22] antitubercular [2,5,7,23], antibacterial [2,5,7,10,11], antimalarial [7,24], anticonvulsant [2,5,24,25], and other types of biological activities [2,5,7,10,11,24] have been identified. 

Recently, the remarkable role of condensed quinazolines in the development and study of novel promising non-steroidal anti-inflammatory drugs (NSAIDs) has been highlighted [24,26,27,28,29,30,31,32,33,34,35,36,37]. Among the diversity of condensed quinazolines (Figure 1), substituted 5,6-dihydrobenzo[*h*]quinazolin-2-amines (**I**) [26] and derivatives of pyrazolo[1,5-*a*]quinazoline (**II**) [27] have been identified as nuclear factor kappa B (NF-κB) inhibitors with anti-inflammatory properties (Figure 1). 2-(4,5-Dihydroxycyclohexa-1,3-dien-1-yl)pyrazolo[5,1-*b*]quinazolin-9(1*H*)-one (**III**) [28] and substituted 2,3-dihydroimidazo[1,2-c]quinazoline-7-carboxamides (**IV**) [29] inhibit aseptic carrageenan-induced inflammation, competing with “Indomethacin” and “Diclofenac”, respectively. *N*-acyl-([1,2,4]triazolo[1,5-*c*]quinazolin-2-yl)alkyl-(alkaryl-, aryl-)amines (**V**) have demonstrated high anti-inflammatory activity in formalin-induced inflammation in rats [30].

Among [1,2,4]triazolo[1,5-*c*]quinazolines with carboxyalkyl substituents at C-2 position (**VI**), compounds with anti-inflammatory effects comparable to the activity of “Sodium diclofenac” in conditions of carrageenan-induced inflammation have been identified [31]. Anti-inflammatory activity is also characteristic of mono- and di-carboxy-containing triazolo[1,5-*c*]quinazolines (**VII**). In this study, it was found that the aforementioned compounds inhibited carrageenan-induced rat paw edema by affecting inflammatory mediators (iNOS activation, decreasing levels of nitrotyrosine, COX-2, and IL-1β) [32]. 2-Benzylidene-3-(R^1^,R^2^-ylidenehydrazineylidene)-2,3,6,7,8,9-hexahydro-5*H*-thiazolo[2,3-*b*]quinazolines (**VIII**) with a 2-hydroxyphenyl substituent at C-5 position also exhibit high anti-inflammatory activity along with a low ulcerogenicity index [33]. It was reported that triazolo[*a*]quinazolines could be effective for the treatment of inflammatory processes, rheumatoid arthritis, gastritis, etc. [24,34]. Furthermore, the effects of 5-chloro-2-methylsulfanyl[1,2,4]triazolo[1,5-*a*]quinazoline (**IX**) on the level of inflammatory mediators (TNF-α, PGE-2) were evaluated. Novel [1,2,4]triazino[2,3-*c*]quinazolines were described as carriers of anti-inflammatory activity (**X–XII**) [35,36,37]. Thus, substituted sodium (3-R^1^-2-oxo-2*H*-[1,2,4]triazino[2,3-*c*]quinazolin-6-yl)alkycarboxylates (**X**), derivatives of 10-R^1^-3-aryl-6,7-dihydro-2*H*-[1,2,4]triazino[2,3-*c*]quinazolin-2-ones of spiro-fused cyclic frameworks (**XI**), and 3-R^1^-2,8-dioxo-7,8-dihydro-2*H*-pyrrolo[1,2-*a*][1,2,4]triazino[2,3-*c*]quinazoline-5*a*(6*H*)carboxylic acids and their salts (**XII**) inhibit carrageenan-induced rat paw edema. Moreover, some of the described compounds surpass sodium diclofenac in terms of anti-inflammatory activity. For example, water-soluble sodium (3-methyl-2-oxo-2*H*-[1,2,4]triazino[2,3-*c*]quinazoline-6-yl)butanoate (compounds **X** if R^1^ = Me, R^2^ = H, n = 2; Figure 1) exceeds the activity of sodium diclofenac (ED_50_: 4 mg/kg compared to 8 mg/kg) [35]. Additionally, the aforementioned compound exhibits antipyretic, antiproliferative, and antisclerogenic effects with low toxicity levels [38,39,40]. It has been shown that compound **XII** is a LOX-15 inhibitor, what could serve as a basis for new directions in the search for innovative anti-inflammatory agents [41]. Bioisosteric replacement is an important instrument in the development of novel non-steroidal anti-inflammatory drugs (NSAIDs) [42,43,44]. It is commonly known that bioisosteric replacement significantly changes the properties of molecules, including their size, shape, electron density distribution, chemical reactivity, lipophilicity, and ability to form hydrogen bonds, among other characteristics. These changes have a significant impact on biological activity, selectivity for biomolecular targets, and toxicity. It should be noted that different chemotypes often exhibit similar patterns of biological activity [45]. In continuation of our previous studies, in the present research, we attempted to improve the anti-inflammatory activity of sodium (3-methyl-2-oxo-2*H*-[1,2,4]triazino[2,3-*c*]quinazoline-6-yl)butanoate (**MTB**) via bioisosteric replacement of a carbon atom with a sulfur atom, additional introduction of substituents to the alkylcarboxylic fragment, and modification of position 3 of the heterocyclic system as a method of lipophilicity modification (Figure 2).

We believe that the introduction of a sulfur atom results in new interactions that stabilize the “ligand–target” complex through additional fixation of the alkylthiocarboxylic fragment and changes the selectivity for biological targets. The presence of a sulfur atom also modifies the lipophilicity of compounds and consequently their pharmacokinetic parameters. Among other effects, it modifies distribution in tissues, particularly in the blood–brain barrier, even if water solubility is decreased. Thus, in the present study, we describe the synthetic approaches to the synthesis of previously described substituted [1,2,4]triazino[2,3-*c*]quinazoline alkylthiocarboxylic acids and their esters as bioisosteric analogs of known anti-inflammatory agents, as well as the evaluation of their pharmacological potential.

## 2. Results and Discussion

### 2.1. Molecular Docking

Molecular docking studies targeting COX-1 and COX-2 were conducted to substantiate the rationale for synthetic studies, optimize further in vivo experiments, and evaluate possible mechanisms of pharmacological activity. Predicted affinity values in kcal/mol and descriptions of interactions with amino acid moieties are shown in Table 1. It was found that almost all studied ligands revealed low predicted affinity toward COX-1, with an affinity range of −2.7 to −6.6 kcal/mol for the studied compounds, while sodium diclofenac showed an affinity of −8.5 kcal/mol. It should be noted that the bioisosteric replacement of carbon by sulfur significantly decreases the calculated affinity toward COX-1; for example, the calculated affinity of MTB is −7.7 kcal/mol. The conformation of the key triazino[2,3-*c*]quinazoline fragment was identical for all studied ligands within the active site of the enzyme. The ligands occupy the position of the phenylacetate moiety of diclofenac (Figure 3b). The ligand molecules are much larger than the diclofenac molecule and therefore form numerous interactions with amino acid moieties that do not belong to the enzyme’s active site and do not participate in the fixation of diclofenac. Furthermore, only for compound **2e**, as well as its analog MTB, interactions were predicted with two key amino acids for enzyme-inhibiting properties, namely, tyrosine (Tyr385) and serine (Ser530) (Appendix A). However, as shown in Figure 3a, these moieties interact with the carbonyl group and triazine ring of compound **2e**, but not with the carboxylic group like classic COX-1 inhibitors [46] and compound MTB. Compound **2e** forms interactions with arginine (Arg120) and tyrosine (Tyr355), which are not experimentally identified moieties of the active site. Thus, COX-1 inhibition by compound **2e** is probable, unlike all other studied compounds. Therefore, it can be concluded that, according to in silico studies, the bioisosteric replacement of a carbon atom by a sulfur atom resulted in a decrease in affinity toward COX-1.

The calculated affinities of the studied ligands toward the COX-2 active site are quite high (−9.4 to −10.8 kcal/mol), though slightly lower than the affinity of the reference drug celecoxib (−12.2 kcal/mol). It should be noted that compound **MTB** also has a high predicted affinity toward COX-2 (−9.9 kcal/mol). The lowest affinity was predicted for compound **2c** (−8.7 kcal/mol), which can be explained by the absence of an interaction between the methylthiopropionic acid moiety and phenylalanine (Phe504) (Figure 4a).

An interaction with this amino acid via a π–sulfur bond was predicted for all other studied ligands (Figure 4b). The formation of the additional π–sulfur bond with Phe504 indicates the relevance of bioisosteric interactions. At the same time, the formation of such a bond is impossible for compound **MTB**, for which anti-inflammatory activity has been proven experimentally (Figure 4c). The compound forms a complex with the active site of the enzyme via a hydrogen bond. Amino acids phenylalanine (Phe504) (Appendix A) and methionine (MET508) are not experimentally identified as part of the active site; hence, their role in the manifestation of anti-inflammatory activity is not obvious. At the same time, the interaction between the methylpropionic acid fragment and glutamine (Gln178) via a hydrogen bond supports the possibility of forming a stable ligand–enzyme complex. Analysis of amino acid interactions reveals the possibility of forming hydrophobic interactions with non-polar amino acid moieties such as valine (Val509, Val335), alanine (Ala513), and leucine (Leu338), which are essential for the formation of stable complexes and the manifestation of the anti-inflammatory activity of coxibs [47]. Additionally, there are hydrogen bonds characteristic of celecoxib with glutamine (Gln178). Visualization of the joint conformations of compound **3b** and celecoxib (the most stable complexes) (Figure 5) reveals a significant similarity in their spatial arrangement within the active site of the enzyme. The obtained data further support the localization of the ligand in the hydrophobic pocket of the COX-2 active site.

### 2.2. Chemistry

6-(Chloro(R^2^)methyl)-3-R^1^-2H-[1,2,4]triazino[2,3-*c*]quinazolin-2-ones [48] (**1.1–1.5**) were used as initial compounds for the synthesis of bioisosteric analogs **2** and **3** (Figure 1). Target products **2a–2i** were obtained by reacting compounds **1a–1e** with corresponding thiol-containing carboxylic acids and their functional derivatives (2-sulfanylacetic acid, 2-sulfanylpropionic acid, ethyl 2-sulfanylacetate, N-acetylcysteine) in propan-2-ol in the presence of sodium hydroxide.

The structure and purity of synthesized compounds **2a–2i**, **3a,** and **3b** were confirmed using ^1^H NMR and HPLC-MS spectra (Appendix A). The signals of molecular ions with *m*/*z* values that correspond to the protonated molecules of synthesized compounds were observed in their HPLC-MS spectra.

In the ^1^H NMR spectra of compounds **2** and **3**, the protons of the triazine ring were registered as an ABCD system consisting of two doublets (H-8 at 7.88–7.82 ppm, H-11 at 8.64–8.57 ppm) and two triplets (H-9 at 8.05–7.92 ppm and H-10 at 7.80–7.64 ppm) [36]. Moreover, in the ^1^H NMR spectra of compounds **2** and **3** signals of protons of substituents at the C-6 position were registered with multiplicities and chemical shifts that correspond to the chemical surroundings. Thus, methylene protons of the -CH_2_SCH_2_COOH fragment in molecules of compounds **2a**, **2g–2i** were registered as two-proton singlets at 4.42–4.34 ppm and 3.48–3.40 ppm, correspondingly. At that time the presence of a methyl group at the *α*- (**2e, 2f**) or γ- (**2b–2d**) position of the abovementioned fragment results in significant changes in the spectral pattern due to additional splitting caused by the effect of an asymmetric carbon atom. Thus, in the ^1^H NMR spectra of compounds **2b–2d,** signals of the -CH_2_SCH(CH_3_)-group protons were registered as two-proton singlet or doublet of doublets at 4.42–4.35 ppm, quadruplet or multiplets at 3.61–3.35 ppm, and doublets at 1.42–1.36 ppm, correspondingly. Spectral patterns of compounds **2e** and **2f** are characterized by the signals of the CH(CH_3_)SCH_2_-group`s protons that were registered in reverse sequence, namely, quadruplet at 5.13–5.04 ppm, doublet of doublets or multiplets at 3.45–3.22 ppm, and doublets at 1.85–1.77 ppm, correspondingly. Signals of the -CH_2_SCH_2_CH(NHAc)-fragment in the ^1^H NMR spectra of compounds **3a** and **3b** are specifically split due to the presence of a chiral center. Signals of protons in the abovementioned fragment were registered as a doublets. The protons of the -COOH-groups of compounds **2a–2c** and **3b** were registered as a singlet at 13.55–11.45 ppm, respectively. In other compounds of this class, the proton carboxylic group was not observed due to the H/D exchange. The ^1^H NMR spectra of esters **2g**–**2i** are characterized by the signals of an ethoxy-group (quadruplet at 3.95–3.93 ppm and triplet at 1.16–1.14 ppm). The signals that correspond to the substituents in position **3** of the heterocyclic system were observed with appropriate chemical shifts and multiplicities [49].

### 2.3. Pharmacology

A toxicity prediction was conducted for synthesized compounds at the initial stage of biological activity screening. This procedure allowed substances for further studies to be selected, and the experimental dosage for in vivo studies was estimated. It was found that the obtained compounds belong to the III–VI class of toxicity and have low predicted side effects (Hepatotoxicity, Carcinogenicity, Immunotoxicity, Mutagenicity, and Cytotoxicity). As shown in Table 2 synthesized compounds are characterized by more auspicious toxicity parameters than the reference compound Diclofenac Sodium.

Considering the satisfactory levels of predicted toxicity and molecular docking results, three compounds were selected for in vivo studies. Compounds **2c** and **2d** were chosen due to their high predicted affinity to the key molecular targets COX-1 and COX-2. Compound **3b**, with a low predicted affinity value, was also selected for an in vivo study to evaluate the possible effect of the binding of the ligand with the phenylalanine (PHE504) moiety on the anti-inflammatory activity of the studied compounds. It should be noted that all synthesized compounds, except **2c**, form a π–sulfur interaction with the phenylalanine (PHE504) moiety, which necessitated the study of its impact on the level of pharmacological activity. Additionally, we included compound **MTB** in the study to verify the reasonableness of using bioisosteric replacement to search for biologically active molecules among the studied series. It was revealed that the studied compounds inhibited the development of the acute phase of inflammation to varying degrees (Table 3). Thus, compound **MTB** inhibited carrageenan-induced paw edema in rats by 45.77%. At the same time, the product of bioisosteric replacement, **2c**, with a methylthiopropionate fragment at the C-6 position of the triazinoquinazoline system, inhibited inflammation by only 3.61%. The replacement of the methylthiopropionate fragment with ethylthioacetate (compound **2e**) resulted in significant anti-inflammatory activity (AA = 53.41%), which is comparable to the effect of the reference drug sodium diclofenac. The introduction of the methylenecysteine fragment was also interesting from the standpoint of improving biological activity (**3b**) (Table 3). Thus, *N*-acetyl-S-((3-(4-i-propylphenyl)-2-oxo-2*H*-[1,2,4]triazino[2,3-*c*]quinazolin-6-yl)methyl)cysteine (**3b**) inhibited the development of paw edema by 51.81% compared with the control group. It should be mentioned that the results of the anti-inflammatory activity study agree with the molecular docking data. Compound **2c**, with low predicted activity toward COX-1 (−4.9 kcal/mol) and COX-2 (−8.7 kcal/mol), shows low anti-inflammatory activity. At the same time, compounds **MTB**, **2e**, and **3b**, with a high predicted affinity toward COX-2, demonstrate high pharmacological effects. Therefore, novel substituted [1,2,4]triazino[2,3-*c*]quinazolinalkylthiocarboxylic acids are promising candidates for further studies aimed at the creation of innovative drugs with anti-inflammatory activity.

The effects of the most active anti-inflammatory agents, **2e** and **3b**, on the level of biochemical markers under conditions of exudative inflammation were studied. Injection of carrageenan into a rat paw led to the development of a typical inflammatory process cascade (Table 4). These processes resulted in an increase in C-reactive protein levels by 6.45 times. C-reactive protein, as a mediator of the acute phase of inflammation, activates the production of cytokines, including interleukin IL-1β, the level of which increased 11.5 times compared to the control group. Additionally, C-reactive protein binds with phospholipids formed as a result of damaged cells, which activates the complement system and subsequent phagocytosis. IL-1β initiates intracellular signal cascades of apoptosis, activates the expression of iNOS, and starts the formation of toxic NO metabolites. These, in turn, enhance nitrosative oxidative stress, initiate intracellular signaling pathways, and promote the expression of pro-inflammatory genes responsible for the synthesis of COX-2 and lipophosphoglycans (LPG). This triggers mechanisms of endothelial dysfunction and apoptosis. Inflammation resulted in an increase in the iNOS level by 9.45 times, leading to subsequent increases in nitrotyrosine and COX-2 levels by 8.37 and 19.29 times, respectively. Also, an increase in VEGF levels by 4.61 times was observed in the blood of animals. VEGF is expressed by IL-1 and COX-2, and its overexpression represents neovascularization in cases of inflammation with a clearly defined inflammatory phase.

The treatment of the studied animals with compounds **2e** and **3b** led to the inhibition of inflammatory processes (Table 3). It has been shown that the studied compounds, along with reducing paw edema, significantly decrease the levels of specific markers that characterize the inflammatory process (COX-2, IL-1β, C-reactive protein), as well as markers of oxidative stress (iNOS, nitrotyrosine) and angiogenesis (VEGF). The most active compound, **2e**, significantly decreased COX-2 expression by 82.5%, IL-1β by 89.5%, and C-reactive protein by 77.1%, confirming its anti-inflammatory properties. Additionally, compound **2e** inhibited the overexpression of VEGF by 62.3%, iNOS by 45.6%, and nitrotyrosine by 70.3%. It should be noted that the effects of compound **2e** on the levels of iNOS and nitrotyrosine exceed those of the reference compound, sodium diclofenac. This finding suggests that compound **2e** has a significant effect on NO-dependent mechanisms involved in the cascade of inflammation development. This is particularly interesting due to the resistance of these mechanisms to pharmacotherapy [50] According to some reports [51], NO-associated mechanisms are central to the development of inflammation and apoptosis [52]. Thus, it has been ascertained that the anti-inflammatory activity of the studied compounds is apparently associated with the disruption of NO-dependent mechanisms, such as the nitrosylation of glutathione and the reduction of glutathione peroxidase activity. The latter controls the synthesis of pro-oxidant metabolites of arachidonic acid, inhibits lipoperoxidation in the cyclooxygenase pathway, and increases COX-2 expression under the action of iNOS [53,54,55]. All of the aforementioned facts support the potential for further study of these compounds.

### 2.4. ADME Prediction

The criteria for “drug-likeness” are crucial in the process of discovering new medicinal agents, providing valuable recommendations in the early stages of drug discovery and increasing the chances of success in clinical trials. These characteristics influence pharmacokinetics (absorption, distribution, metabolism, and excretion) and, consequently, the pharmacological activity and efficacy of the investigated medicinal agents. The “drug-likeness” criteria for the compounds **2e**, **3b**, and **MTB** are presented in Table 5.

Compound **2c**, as well as MTB, complies with the drug-likeness criteria: MW (Da) (<500), n-HBA (<10), n-HBD (≤5), TPSA (<140 Å^2^), and logP (≤5). The satisfactory value of TPSA (>140 Å^2^) for compound **2e** correlates well with passive molecular transport through membranes. Compound **2e** has a high ability to pass through the blood–brain barrier and flexibly interact with molecular targets. The studied compound reveals high predicted bioavailability (0.56). Moreover, compound **2e**, unlike compound **3b**, passes all commonly used filters (Lipinski, Veber, Muegge, Ghose, Egan). Additionally, compound **2e** is lead-like and suitable for further modification.

## 3. Materials and Methods

### 3.1. Chemistry

Melting points were determined in open capillary tubes in a «Mettler Toledo MP 50» (Columbus, OH, USA) apparatus and were uncorrected. The elemental analyses (C, H, N) were performed using the ELEMENTAR vario EL cube analyzer (Langenselbold, Germany). Analyses were indicated by symbols of the elements or functions within ±0.3% of the theoretical values. ^1^H NMR spectra (500 MHz) were recorded on a Varian Mercury 500 (Varian Inc., Palo Alto, USA) spectrometers with TMS as an internal standard in DMSO-d_6_ solution. LC-MS was recorded using a chromatography/mass spectrometric system which consists of high-performance liquid chromatography «Agilent 1100 Series» (Agilent, Palo Alto, CA, USA) equipped with diode-matrix and mass-selective detector «Agilent LC/MSD SL» (Agilent, Palo Alto, CA, USA) (atmospheric pressure chemical ionization—APCI). The purity of all obtained compounds was checked by ^1^H-NMR and LC-MS.

General procedure for the synthesis of 2-[((3-R-2-oxo-2*H*-[1,2,4]triazino[2,3-*c*]quinazolin-6-yl)methyl)thio]carboxylic acid (**2a**–**2f**).

Briefly, 25 mM of the corresponding sulfanylcarboxylic acid was placed in a round bottom flask, 5 mL of 1 M sodium hydroxide solution (50 mM) and 25 mL of propan-2-ol were added, stirred, and left for 2 min. Then, 25 mM of the starting substance (**1**) was added to the prepared solution and refluxed for 2 h. The reaction mixture was cooled, acidified with 1M hydrochloric acid to pH 4, and the precipitate was filtered off and dried.

2-(((2-oxo-3-phenyl-2*H*-[1,2,4]triazino[2,3-*c*]quinazolin-6-yl)methyl)thio)acetic acid (**2a**).

Yield: 74%; M.p. 227–229 °C; ^1^H NMR (500 MHz, DMSO-d_6_) δ 11.45 (s, 1H, COOH), 8.63 (d, *J* = 7.8 Hz, 1H, H-11), 8.33 (d, *J* = 6.8 Hz, 2H, 3 Ar H-2,6), 7.97 (t, *J* = 7.6 Hz, 1H, H-9), 7.87 (d, *J* = 7.9 Hz, 1H, H-8), 7.74 (t, *J* = 7.5 Hz, 1H, H-10), 7.64–7.49 (m, 3H, 3 Ar H-3,4,5), 4.37 (s, 2H, -CH_2_S-), 3.40 (s, 2H, -SCH_2_-); LC-MS (*m*/*z*) = 379.0 (M + H^+^)^+^; Calculated for C_19_H_14_N_4_O_3_S: C, 60.31; H, 3.73; N, 14.81; S, 8.47; Found: C, 60.35; H, 3.75; N, 14.83; S, 8.45.

2-(((2-oxo-3-phenyl-2*H*-[1,2,4]triazino[2,3-*c*]quinazolin-6-yl)methyl)thio)propanoic acid (**2b**).

Yield: 54%; M.p. 222–224 °C; ^1^H NMR (500 MHz, DMSO-d_6_) δ 12.53 (s, 1H, COOH), 8.63 (d, *J* = 7.4 Hz, 1H, H-11), 8.33 (d, *J* = 6.8 Hz, 2H, 3 Ar H-2,6), 8.05–7.93 (m, 1H, H-9), 7.87 (d, *J* = 8.0 Hz, 1H, H-8), 7.74 (t, *J* = 7.5 Hz, 1H, H-10), 7.63–7.45 (m, 3H, 3 Ar H-3,4,5), 4.42 (s, 2H, -CH2S-), 3.61 (q, *J* = 7.1 Hz, 1H, -SCH(CH_3_)-), 1.42 (d, *J* = 7.2 Hz, 3H, -SCH(CH_3_)-); LC-MS (*m*/*z*) = 393.1 (M + H^+^)^+^; Calculated for C_20_H_16_N_4_O_3_S: C, 61.21; H, 4.11; N, 14.28; S, 8.17; Found: C, 61.20; H, 4.07; N, 14.31; S, 8.21

2-(((3-(4-i-Propylphenyl)-2-oxo-2*H*-[1,2,4]triazino[2,3-*c*]quinazolin-6-yl)methyl)thio)propanoic acid (**2c**).

Yield: 49%; M.p. 206–207 °C; ^1^H NMR (500 MHz, DMSO-d_6_) δ 12.37 (s, 1H, COOH), 8.59 (d, *J* = 8.0 Hz, 1H, H-11), 8.25 (d, *J* = 8.3 Hz, 2H, 3-Ar H-2,6), 7.94 (t, *J* = 8.2 Hz, 1H, H-9), 7.84 (d, *J* = 8.0 Hz, 1H, H-8), 7.70 (t, *J* = 7.6 Hz, 1H, H-10), 7.35 (d, *J* = 8.3 Hz, 2H, 3-Ar H-3,5), 4.38 (s, 2H, -CH_2_S-), 3.58 (q, *J* = 7.2 Hz, 1H, -SCH(CH_3_)-), 2.99 (dt, *J* = 13.7, 6.8 Hz, 1H, -CH(CH_3_)_2_), 1.40 (d, *J* = 7.2 Hz, 3H, -SCH(CH_3_)-), 1.29 (d, *J* = 6.9 Hz, 6H, -CH(CH_3_)_2_).; LC-MS (*m*/*z*) = 435.0 (M + H^+^)^+^; Calculated for C_23_H_22_N_4_O_3_S: C, 63.58; H, 5.10; N, 12.89; S, 7.38; Found: C, 63.62; H, 5.09; N, 12.93; S, 7.35.

2-(((3-(4-Fluorophenyl)-2-oxo-2*H*-[1,2,4]triazino[2,3-*c*]quinazolin-6-yl)methyl)thio)propanoic acid (**2d**).

Yield: 50%; M.p. 235–236 °C; ^1^H NMR (500 MHz, DMSO-d_6_) δ 8.61 (d, *J* = 8.1 Hz, 1H, H-11), 8.45 (dd, *J* = 8.3, 5.9 Hz, 2H, 3-Ar H-2,6), 8.01–7.92 (m, 1H, H-9), 7.88 (d, *J* = 8.3 Hz, 1H, H-8), 7.79–7.64 (m, 1H, H-10), 7.27 (t, *J* = 8.7 Hz, 2H, 3-Ar H-3,5), 4.35 (dd, *J* = 58.9, 13.9 Hz, 2H, -CH_2_S-), 3.51–3.35 (m, 1H, -SCH(CH_3_)-), 1.36 (d, *J* = 7.1 Hz, 3H, -SCH(CH_3_)-. LC-MS (*m*/*z*) = 411.0 (M + H^+^)^+^; Calculated for C_20_H_15_FN_4_O_3_S: C, 58.53; H, 3.68; N, 13.65; S, 7.81 Found: C, 58.55; H, 3.71; N, 13.67; S, 7.83.

2-((1-(3-Methyl-2-oxo-2*H*-[1,2,4]triazino[2,3-*c*]quinazolin-6-yl)ethyl)thio)acetic acid (**2e**).

Yield: 71%; M.p. 260–262 °C; ^1^H NMR (500 MHz, DMSO-d_6_) δ 8.57 (d, *J* = 8.1 Hz, 1H, H-11), 7.92 (t, *J* = 7.6 Hz, 1H, H-9), 7.82 (d, *J* = 8.1 Hz, 1H, H-8), 7.68 (t, *J* = 8.0 Hz, 1H, H-10), 5.04 (q, *J* = 6.8 Hz, 1H, -CH(CH_3_)S-), 3.30 (dd, *J* = 42.1, 15.7 Hz, 2H, -SCH_2_-), 2.43 (s, 3H, -CH_3_), 1.77 (d, *J* = 7.0 Hz, 3H, -CH(CH_3_)S-); LC-MS (*m*/*z*) = 331.0 (M + H^+^)^+^; Calculated for C_15_H_14_N_4_O_3_S: C, 54.54; H, 4.27; N, 16.96; S, 9.70; Found: C, 54.57; H, 4.29; N, 17.01; S, 9.72.

2-((1-(2-oxo-3-phenyl-2*H*-[1,2,4]triazino[2,3-*c*]quinazolin-6-yl)ethyl)thio)acetic acid (**2f**).

Yield: 45%; M.p. 213–215 °C; ^1^H NMR (500 MHz, DMSO-d_6_) δ 8.62 (d, *J* = 7.3 Hz, 1H, H-11), 8.34 (d, *J* = 6.6 Hz, 2H, 3 Ar H-2,6), 7.97 (t, *J* = 7.1 Hz, 1H, H-9), 7.87 (d, *J* = 8.0 Hz, 1H, H-8), 7.73 (t, *J* = 7.2 Hz, 1H, H-10), 7.55 (t, *J* = 7.5 Hz, 3H, 3 Ar H-3,4,5), 5.13 (q, *J* = 6.1 Hz, 1H, -CH(CH_3_)S-), 3.45–3.22 (m, 2H, -SCH_2_-), 1.85 (d, *J* = 6.9 Hz, 3H, -CH(CH_3_)S-); LC-MS (*m*/*z*) = 393.0 (M + H^+^)^+^; Calculated for C_20_H_16_N_4_O_3_S: C, 61.21; H, 4.11; N, 14.28; S, 8.17; Found: C, 61.20; H, 4.15; N, 14.31; S, 8.20.

General procedure for the synthesis of 2-[((3-R-2-oxo-2H-[1,2,4]triazino[2,3-*c*]quinazolin-6-yl)methyl)thio]acetates (**2g**–**2i**).

Briefly, 0.3 g (25 mM) ethyl sulfanylacetate was placed in a round bottom flask, 2.5 mL of 1 M sodium hydroxide solution (25 mM) and 20 mL of propan-2-ol were added, stirred, and left for 2 min. Then, 25 mM of the starting substances (**1**) was added to the prepared solution and was refluxed for 2 h. The reaction mass was cooled. The precipitate that was formed was filtered off and dried.

Ethyl 2-(((2-oxo-3-phenyl-2*H*-[1,2,4]triazino[2,3-*c*]quinazolin-6-yl)methyl)thio)acetate (**2g**).

Yield: 65%; M.p. 220–222 °C; ^1^H NMR (500 MHz, DMSO-d_6_) δ 8.64 (d, *J* = 7.2 Hz, 1H, H-11), 8.32 (d, *J* = 6.7 Hz, 2H, 3 Ar H-2,6), 7.97 (t, 1H, H-9), 7.85 (d, *J* = 7.9 Hz, 1H, H-8), 7.75 (t, *J* = 7.5 Hz, 1H, H-10), 7.63–7.38 (m, 3H, 3 Ar H-3,4,5), 4.37 (s, 2H, -CH_2_S), 3.95 (q, *J* = 7.1 Hz, 2H, -OCH_2_CH_3_), 3.48 (s, 2H, -SCH_2_--), 1.16 (t, *J* = 7.1 Hz, 3H, -OCH_2_CH_3_); LC-MS (*m*/*z*) = 407 (M + H^+^)^+^; Calculated for C_21_H_18_N_4_O_3_S: C, 62.06; H, 4.46; N, 13.78; S, 7.89; Found: C, 62.08; H, 4.49; N, 13.81; S, 7.90.

Ethyl 2-(((3-(4-i-propylphenyl)-2-oxo-2*H*-[1,2,4]triazino[2,3-*c*]quinazolin-6-yl)methyl)thio)acetate (**2h**).

Yield: 62%; M.p. 150–151 °C; ^1^H NMR (500 MHz, DMSO-d_6_) δ 8.61 (d, *J* = 7.9 Hz, 1H, H-11 Hz), 8.24 (d, *J* = 8.2 Hz, 2H, 3-Ar H-2,6), 7.94 (t, *J* = 7.6 Hz, 1H, H-9), 7.82 (d, *J* = 8.1 Hz, 1H, H-8), 7.72 (t, *J* = 7.3 Hz, 1H, H-10), 7.35 (d, *J* = 8.2 Hz, 2H, 3-Ar H-3,5), 4.34 (s, 2H, -CH_2_S-), 3.93 (q, *J* = 7.1 Hz, 2H, -OCH_2_CH_3_), 3.45 (s, 2H, -SCH_2_-), 3.01–2.93 (m, 1H, -CH(CH_3_)_2_), 1.30 (d, *J* = 6.8 Hz, 6H, -CH(CH_3_)_2_), 1.16 (t, *J* = 7.1 Hz, 3H, -OCH_2_CH_3_); LC-MS (*m*/*z*) = 449.0 (M + H^+^)^+^; Calculated for C_24_H_24_N_4_O_3_S: C, 64.27; H, 5.39; N, 12.49; S, 7.15; Found: C, 64.24; H, 5.38; N, 12.54; S, 7.14.

Ethyl 2-(((3-(4-fluorophenyl)-2-oxo-2*H*-[1,2,4]triazino[2,3-*c*]quinazolin-6-yl)methyl)thio)acetate (**2i**).

Yield: 90%; M.p. 149–151 °C; ^1^H NMR (500 MHz, DMSO-d6) δ 8.62 (d, *J* = 8.0 Hz, 1H, H-11), 8.42 (dd, *J* = 8.6, 5.7 Hz, 2H, 3-Ar H-2,6), 7.95 (t, *J* = 7.6 Hz, 1H, H-9), 7.83 (d, *J* = 8.1 Hz, 1H, H-8), 7.73 (t, *J* = 7.6 Hz, 1H, H-10), 7.25 (t, *J* = 8.7 Hz, 2H, 3-Ar H-3,5), 4.35 (s, 2H, -CH_2_S-), 3.92 (q, *J* = 7.1 Hz, 2H, -OCH_2_CH_3_), 3.45 (s, 2H, -SCH_2_-), 1.14 (t, *J* = 7.1 Hz, 3H, -OCH_2_CH_3_); LC-MS (*m*/*z*) = 425.0 (M + H^+^)^+^; Calculated for C_21_H_17_FN_4_O_3_S: C, 59.43; H, 4.04; N, 13.20; S, 7.55; Found: C, 59.45; H, 4.05; N, 13.25; S, 7.57.

General procedure for the synthesis of *N*-acetyl-S-((2-oxo-3-R-2*H*-[1,2,4]triazino[2,3-*c*]quinazolin-6-yl)methyl)cysteines (**3a, 3b**).

Briefly, 0.408 g (25 mM) of acetylcysteine was placed in a round bottom flask, 5 mL of 1 M sodium hydroxide solution (50 mM) and 20 mL of propan-2-ol, stirred and left for 2 min. Then, 25 mM of the corresponding starting substances (**1**) was added to the prepared solution and was refluxed for 2 h. The reaction mixture was cooled, acidified with 1M hydrochloric acid to pH 4, the precipitate formed was filtered off, and then washed with water and alcohol and dried.

*N*-acetyl-S-((2-oxo-3-phenyl-2*H*-[1,2,4]triazino[2,3-*c*]quinazolin-6-yl)methyl)cysteine (**3a**).

Yield: 72%; M.p. 238–239 °C; ^1^H NMR (500 MHz, DMSO-d_6_) δ 8.60 (d, *J* = 7.9 Hz, 1H, H-11), 8.31 (d, *J* = 7.8 Hz, 2H, 3-Ar H-2,6), 8.03–7.92 (m, 1H, H-9), 7.88 (s, 1H, -NHAc), 7.80–7.65 (m, 1H, H-10), 7.61–7.43 (m, 3H, 3-Ar H-3,4,5), 7.27 (d, *J* = 6.9 Hz, 1H, H8), 4.24 (s, 2H, -CH_2_S-), 4.16–4.00 (m, 1H, -SCH_2_CH(NHAc)-), 3.26–2.91 (m, 2H, -SCH_2_CH(NHAc)-), 1.77 (s, 3H, -COCH_3_). LC-MS (*m*/*z*) = 450.0 (M + H^+^)^+^; Calculated for C_22_H_19_N_5_O_4_S: C, 58.79; H, 4.26; N, 15.58; S, 7.13; Found: C, 58.82; H, 4.27; N, 15.63; S, 7.15.

*N*-acetyl-S-((3-(4-i-propylphenyl)-2-oxo-2H-[1,2,4]triazino[2,3-c]quinazolin-6-yl)methyl)–cysteine (**3b**).

Yield: 89%; M.p. 190–191 °C; ^1^H NMR (500 MHz, DMSO-d_6_) δ 13.55 (s, 1H, COOH), 8.57 (d, *J* = 8.0 Hz, 1H, H-11), 8.24 (d, *J* = 8.2 Hz, 2H, 3-Ar H-2,6), 7.91 (t, *J* = 7.4 Hz, 1H, H-9), 7.86 (d, *J* = 7.9 Hz, 1H, H-8), 7.68 (t, *J* = 7.5 Hz, 1H, H-10), 7.55 (d, *J* = 7.1 Hz, 1H, -NHAc), 7.35 (d, *J* = 8.2 Hz, 2H, 3-Ar H-3, 5), 4.37–4.05 (m, 3H, -CH_2_SCH_2_CH(NHAc)-), 3.16–3.06 (m, 2H, -SCH_2_CH(NHAc)-), 3.04–2.86 (m, 1H, -CH(CH_3_)_2_), 1.78 (s, 3H, -COCH_3_), 1.29 (d, *J* = 6.9 Hz, 6H, -CH(CH_3_)_2_); LC-MS (*m*/*z*) = 492.0 (M + H^+^)^+^; Calculated for C_25_H_25_N_5_O_4_S: C, 61.09; H, 5.13; N, 14.25; S, 6.52; Found: C, 61.07; H, 5.10; N, 14.27; S, 6.50.

### 3.2. Pharmacological/Biological Assays

#### 3.2.1. Anti-Inflammatory Studies

Animals. Adult Wistar white rats (weighing 150–160 g) were attained from the animal house facility of the «Institute of Pharmacology and Toxicology of Ukraine» (Kyiv) and were handled under controlled temperature (25 °C ± 3 °C), humidity (50–60%), and illumination (12 h light and 12 h dark cycles, lights on at 08:00) conditions. Before starting the experiment, the rats were given water ad libitum and the standard pellet diet for 1 week. All experimental procedures and treatments were carried out according to the European Convention and «Regulations on the use of animals in biomedical research» [56].

Anti-inflammatory activity. Screening of synthesized compounds with estimated anti-inflammatory activity began with the study of their effect on the exudative phase of acute aseptic inflammation («carrageenan» test). Phlogogen (1% aqueous solution of carrageenan (Sigma, St. Louis, MO, USA)) was subplantally injected in a dose of 0.1 mL in the rats’ back right paw [57]. The left one was used as a control. Intragastric administration of studied compounds was conducted using an atraumatic probe as water solution or finely dispersed suspension stabilized by Tween—80 in a dose of 25 mg/kg 1 h before the injection of phlogogen. The reference drug «Diclofenac sodium» was administered intragastrically in a recommended dose of 8 mg/kg for pre-clinical studies. Measurements of paw volume were conducted before the experiment and at 3 h after injection of phlogogen using the described methods. The activity of these substances was determined by their ability to reduce the swelling compared with control group and was expressed in percentage. Using a water displacement plethysmometer (Ugo Basile, Comerio, Italy), paw volume was measured. It showed how the substance inhibited carrageenan swelling in relation to control swelling where the value was taken as 100%. The activity of the studied compounds was calculated as follows (1):A = 100% − ((Vse − Vhe)/(Vsc − Vhc) × 100%),(1)
where A—anti-exudative activity, %; Vse—the volume of swollen paw in the experiment; Vhe—the volume of healthy paw in the experiment; Vsc—the volume of swollen paw in control; Vhc—the volume of healthy paw in control.

#### 3.2.2. Study of Molecular Markers of Inflammation

At the end of the experiment, 3 h after the last dose of treatment, blood samples were collected from the tail vein with a special syringe with a heparinized needle. Blood samples were centrifuged for 15 min at 2500 rpm, and “sa53q 11” was used to determine the levels of C-reactive protein, intelleukin-1b (IL-1β), inducible NO-synthase (iNOS), cyclooxygenase-2 (COX-2), vasculoendothelial growth factor (VEGF), and nitrotyrosine. Content of C-reactive protein was evaluated by solid-phase immunosorbent sandwich method ELISA, ELISA Kit (Biomerica, Irvine, CA, USA, ReF: 7033, Lot 2349) and expressed in ng/l. Interleukin-1b (IL-1β) was evaluated by solid-phase immunosorbent sandwich method ELISA, ELISA Kit (Bioscience, London, UK, ReF: BMS224 HS, Lot 105505000) and expressed in pg/mL. Inducible NO-synthase (iNOS) was evaluated by solid-phase immunosorbent sandwich method ELISA, (Elabscience, Houston, TX, USA)—enzyme immunoassay (analyzer Immunochem-200 (HTI, North Attleboro, MA, USA) ELISA Kit (E-EL-R0520) and expressed in pg/mL. Cyclooxygenase-2 (COX2) was evaluated by solid-phase immunosorbent sandwich method ELISA, (Cloud-Clone Corporation, Katy, TX, USA)—enzyme immunoassay (analyzer—Immunochem-200 (HTI, North Attleboro, USA) ELISA Kit (SEA699Ra) and expressed in pg/mL. Nitrotyrosine—was evaluated by solid phase enzyme immunoassay (ELISA) on a full-plate enzyme immunoassay analyzer (SIRIO S, Pomezia, Italy) with the application of test kit «Nitrotirosine ELISA Kit»(«HyCult biotechnology» (Uden, The Netherlands) ELISA Kit HK501-02) and expressed in nM/mL. Vascular endothelial growth factor—was evaluated by solid-phase enzyme immunoassay (ELISA) on a full-plate enzyme immunoassay analyzer (SIRIO S, Pomezia, Italy) with the application of test kit Rat VEGF ELISA Kit is a sandwich (quantitative) ELISA abcam No. ab 100738 and expressed in pg/mL. Statistical data processing was performed using a license program «STATISTICA^®^ for Windows 6.0» (StatSoftInc., Tulsa, OK, USA, No. AXXR712D833214FAN5) and «SPSS 16.0», «Microsoft Office Excel 2003». The results were presented as mean ± standard error of the mean. Arithmetic mean and standard error of the mean were calculated for each of the studied parameters. During the verification of the statistical hypothesis, the null hypothesis was declined if the statistical criterion was *p* < 0.05 [58].

### 3.3. Molecular Docking

Research was conducted by flexible molecular docking, as an approach of finding molecules with affinity to a specific biological target. Macromolecules from Protein Data Bank (PDB) were used as biological targets, namely, COX-1 enzyme in complex with diclofenac (PDB ID—3N8Y) [46] and COX-2 enzyme in complex with celecoxib (PDB ID—3LN1) [47]. The software used and the molecular docking parameters are described earlier. Validation of the native ligand docking methodology, and the calculation of RMSD between the reference and native conformations of diclofenac and celecoxib, respectively, were also described previously [32].

### 3.4. Toxicity Prediction

The ProTox-II site (https://tox-new.charite.de/protox_II/index.php?site=compound_input, accessed on 21 May 2024) was used to predict the toxicity criteria of molecules [59]. It incorporates molecular similarity, fragment propensities, and machine learning, based on a total of 33 models for the prediction of various toxicity endpoints such as acute toxicity, hepatotoxicity, cytotoxicity, carcinogenicity, mutagenicity, immunotoxicity, adverse outcomes (Tox21) pathways, and toxicity targets.

### 3.5. SwissADME Prediction

The SwissADME site was used to calculate physicochemical descriptors, as well as to predict ADME parameters, pharmacokinetic properties, and drug similarity. The basic approaches and basic methodology of SwissADME, as a free web-based tool for evaluating pharmacokinetics and drug-likeness, are described in recent publications [60,61,62].

## 4. Conclusions

Bioisosteric replacement of carbon by sulfur proved to be an efficient approach for the purposeful search of anti-inflammatory agents among carboxyl-containing [1,2,4]triazino[2,3-*c*]quinazolines. 2-[((3-R-2-oxo-2*H*-[1,2,4]triazino[2,3-*c*]quinazolin-6-yl)methyl)thio]alkylcarboxylic acids, designed according to the abovementioned approach, are available through the interaction of 6-(chloro(R^2^)methyl)-3-R^1^-2*H*-[1,2,4]triazino[2,3-i]quinazolin-2-ones with sulfanylcarboxylic acids. Molecular docking studies revealed the role of the introduced sulfur atom in the formation of binding between the ligand and the biomolecular target complex. According to the obtained results, the sulfur-containing compounds form an additional π–sulfur interaction which, as we consider, enhances their anti-inflammatory properties. According to in vivo studies, compounds **2e** and **3b** reveal anti-inflammatory activity comparable with the pharmacological effect of the reference compound sodium diclofenac. Additionally, the studied compounds significantly modify the content of inflammation markers and mediators in the serum of experimental animals. The established inhibition of COX-2 expression by the obtained compounds agrees with the results of docking studies, which additionally supports COX-2 inhibition as a probable mechanism of their anti-inflammatory activity. Compound **2e** was identified as lead-like and suitable for further modification aimed at the creation of innovative anti-inflammatory agents.

## Data Availability

All the data generated during this research are included in the manuscript.

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
