# Peer review of "Bioisosteric Replacement in the Search for Biologically Active Compounds: Design, Synthesis and Anti-Inflammatory Activity of Novel [1,2,4]triazino[2,3-c]quinazolines"

_pharmaceuticals, 2024, doi:10.3390/ph17111437_

Round 1

Reviewer 1 Report

Comments and Suggestions for Authors

The manuscript prepared by Oleksandr Grytsak and co-workers describes the design and synthesis of [1,2,4]triazino[2,3-c]quinazoline derivatives as novel anti-inflammatory agents. The compounds presented were prepared according to the docking studies discussed in the paper. The topic of this manuscript could be important and of interest to researchers involved in the development of new drug-like compounds exhibiting anti-inflammatory activity and/or in the synthesis of quinazoline-based compounds. In my opinion, the presentation of the obtained results should be clearer and more comprehensible to make the work seem interesting to the reader. In addition, the manuscript contains a number of formal deficiencies that should be eliminated (please see my comments and recommendations in the attached file). Overall, the work is of average novelty and quality.

Despite all the comments mentioned in my review, I believe that the manuscript could be accepted for publication in Pharmaceuticals after major revision.

Author Response

We thank the reviewer for the evaluation and constructive feedback.

Reviewer 1.

Reviewer 1 comment 1.

I suggest replacing the term “mercapto” (line 34, 201, 202, 382, 436) with the term “sulfanyl” (according to the IUPAC Golden Book recommendation)

b) I suggest replacing term “propanol-2 with “propan-2-ol” (line 202, 383, 437, 469)

c) I ask the authors to check the nomenclature used throughout the manuscript, especially the use of italics in compound names (e.g., line 31, 32, 34, 67, 72, 82, 92, 285, 286, 292, 573)

d) I ask the authors to consider replacing the term “Pi-Sulfur” with “π-sulfur”

Authors response 1. We made all recommended corrections.

Reviewer 1 comment 2.

The wording “in the 2nd position” (line 75), “in the 5th position” (line 83) or “in 6th position” (line 219) could be confusing for the reader. I recommend using “at C-5 position”, for example.

Authors response 2. We made all recommended corrections.

Reviewer 1 comment 3.

The compound listed in lines 95 and 96 should not be referred to as “compound X” because it is clear from Figure 1 that there are several chemical entities represented by “X”.

Authors response 3. We added the information about nature of substituents to clarify the nature of compound.

Reviewer 1 comment 4.

MTB is listed both as an acid (line 109) and its sodium salt (Figure 2); please unify.

Authors response 4. We unified the information about MTB nature.

Reviewer 1 comment 5.

Table 1 is too extensive and difficult to read. I ask the authors to:

a) state only the values of the affinity for COX-1/2 in Table 1

b) move columns 3 a 5 (“Amino acid moiety and bonding type”) to a new Table S1 and place this table in the Supporting Information file (see my detailed comment bellow; point 13)

c) add a title to the table

Authors response 5. We edited Table 1 and moved information about nature of Amino acid moiety and bonding type to the Supporting Information file.

Reviewer 1 comment 6.

Scheme 1 is confusing at several points:

a) The illustration of R, R1 and R2 substituents in not consistent with Figure 2. I ask the authors to unify.

b) I strongly urge the authors to renumber of the compounds presented in the work (e.g., 1a–1e instead of 1.1–1.5, etc.). The currently used compound numbering is confusing (it is the same as the chapter numbering).

c) It is not possible to assign the structure of the presented compounds from Scheme 1 (e.g., what is the structure of compound 2.5?). I recommend adding a key for the substituents.

d) The structures of the presented compounds could be clearer. In the case of compounds bearing a stereogenic centre (e.g., the C—NHAc bond of compounds 3.1 and 3.2), this should be shown as a wavy line to clarify that the compound was not prepared in pure enantiomeric form, but as a racemate. If I am not mistaken, this note refers to seven compounds.

Authors response 6. We edited Scheme 1 according to the recommendation of the Reviewer.

Reviewer 1 comment 7.

The paragraph describing the results from 1H NMR spectroscopy is difficult to read and a little confusing. I suggest:

a) adding the numbering of H atoms from the triazine ring to Figure 2 or Scheme 1

b) including one specific example of the 1H NMR spectra in the manuscript with a detailed description (e.g., for compound 2.5)

c) Moreover, I would like to ask the authors if they observed one or two sets of signals in the 1H NMR spectra of compounds containing a stereogenic centre(?)

Authors response 7. We added information about atoms` numbering to the Scheme 1. We considered all obtained compounds as racemates because we used racemic reagents. No new asymmetric center was formed as result of synthetic procedures made by us. While planning the study we avoided the presence of two asymmetric carbons in one molecule. We observed one set of protons in 1H NMR spectra of obtained compounds. We ask the reviewer to allow leaving unchanged fragment devoted to the NMR spectra description.

Reviewer 1 comment 8.

The authors claim to have seen “signal of molecular ions…” (line 205) in the mass spectra of prepared compounds. This seems unlikely given that an APCI ion source generating predominantly ions that can be assigned as “protonated molecules” (M+H+)+ was used. What type of ions did the authors observe in the mass spectra?

Authors response 8.

For most of obtained compounds we used both positive and negative polarization, thus we observed as positive so negative ions. However we decided to include to manuscript only information about (M+H+)+. We replaced “signal of molecular ions…” by “signal of protonated molecules ions…” in the text.

Reviewer 1 comment 9.

The interpretation of LC-MS results reported in Materials and Methods (Chemistry) should be improved:

a) results should be given to one decimal place (if a low-resolution mass spectrometer was used)

b) the notation of the observed ion format should be changed to “(M+H+)+” instead of “(M+H)”

c) for compounds 2.7, 3.1 and 3.2, the specification of the observed ion is completely missing

Authors response 9. We made recommended corrections.

Reviewer 1 comment 10.

The titles of Tables 1, 2 and 4 are not provided in the manuscript.

Authors response 10. We added titles of Tables 1, 2 and 4 to the manuscript.

Reviewer 1 comment 11.

Lines 250–255 and 266–271 contain the same text.

Authors response 11. The duplicate of the text was removed.

Reviewer 1 comment 12. Figures 3b and 4a are identical.

Authors response 12. Figure 4a was replaced by correct image.

Reviewer 1 comment 13.

The authors did not provide a Supporting Information file. I ask the authors to prepare this file containing:

a) detailed parameters of LC-MS measurement conditions (for both LC and MS)

b) 1H NMR spectra of prepared compounds with signal assignments

c) APCI-MS with signal assignments

d) Table S1 with detailed results from docking experiments

Authors response 13.

The Supporting Information was prepared.

Reviewer 1 comment 14.

Some typographical errors can be found in the manuscript:

a) the orientation of R1 and R2 substituent in compounds X, XI and XII (Figure 1) should be the same

b) the use of “ED50” is more usual than “ED50”

c) the frame in Figure 2 overlaps the letter “H” from COOH group

d) I suggest using italics for some terms, such as “in vivo” (line 132, 251, 269, 580)

e) in many cases, the authors use a comma instead of a period in the decimal point (e.g., Table 1 (MTB), Table 2, Table 3 and Table 4)

f) Table 3 errors: i) column 1 (“MTB” instead of “MTB#“, “DS*#” instead of “DS#*(as mentioned in “notes” under the table); ii) column 3 contains in title “4h h of…” which seems to be a typographical error

g) Table 4 errors: i) column 1 (“IL-1b” instead of “IL-1β”); ii) there is a different font size and style in the last column (MTB); iii) in column 4 (last line), I do not understand the meaning of the symbol “*##“; iv) the use of different concentration units may be a bit confusing for the reader

h) Table 5 errors: i) what does “compounds 2” mean in the table title?; ii) in column 1 (line 4 and 6), the size and style of the symbol “≤” should be changed

i) the hyphen in the words “com-pound” (line 584) and “crea-tion” (line 587) should be deleted

Authors response 14.

The recommended corrections were made by the authors. The only exception was comment about different concentration units. In this manuscript we indicated units that commonly used for each proper marker.

Reviewer 2 Report

Comments and Suggestions for Authors

attached 

Author Response

We thank the reviewer for the evaluation and constructive feedback.

Q 1.The abstract lacks specific quantitative data obtained from different experiments. Please add some quantitative data to the abstract.

Authors response 1. We added quantitative data about anti-inflammatory activity of most active compounds.

Q 2.

The work is mainly based on prediction models. I would like to test the validity of these predictions with multiple models, not just one. For example, docking should be tested in more than one program. The same is true for any part of the workflow done computationally.

Authors response 2. We agree with the reviewer that there is an option of cross-validating docking methodology with different programs, just as molecular docking can also be performed by several programs to compare the results. However, the presented docking methodology using diclofenac (PDB ID - 3N8Y) and celecoxib (PDB ID - 3LN1) as reference ligands has been used by us repeatedly, and the correlation between in silico and in vivo experimental results has been proven earlier (Krasovska, N. ; Berest, G.; Belenichev, I.; Severina, H.; Nosulenko, I.; Vosokoboinik, O.; Okovytyy, S.; Kovalenko, S. 5+1-Heterocyclization as a preparative approach for carboxy-containing triazolo[1,5-c]quinazolines with anti-inflammatory activity. European Journal of Medicinal Chemistry. 2024, 266, 116137. https://doi.org/10.1016/j.ejmech.2024.116137; Citation: Sepp, J.; Koshovyi, O.; Jakštas, V.; Žvikas, V.; Botsula, I.; Kireyev, I.; Severina, H.; Kukhtenko, O.; Põhako-Palu, K.; Kogermann, K.; et al. Phytochemical, Pharmacological, and Molecular Docking Study of Dry Extracts of Matricaria discoidea DC. with Analgesic and Soporific Activities. Biomolecules 2024, 14, 361. https://doi.org/10.3390/biom14030361). In the presented manuscript, we also observe the correspondence of these results. Therefore, in our opinion, this allows us to limit the use of one program and not to overload the manuscript with information on additional validation characteristics and affinity parameters.

Q 3.

The carrageenan-induced paw edema model used for testing the anti-inflammatory effect primarily measures acute inflammation, typically over a short period (4-6 hours). It does not provide insights into chronic inflammatory conditions, often more relevant for diseases such as arthritis or inflammatory bowel disease.

Authors response 3. The study of the anti-inflammatory activity in chronic inflammatory conditions is element of the in-depth study of promising anti-inflammatory agents. Our paper devoted to the evaluation of the reasonableness of bioisosteric replacement as part of the strategy of development of anti-inflammatory drugs. Thus, we did not study activity into chronic inflammatory conditions.

Q 4. There is limited discussion about the scalability of the synthetic methods used, which may be a concern for further developing these compounds as drug candidates.

Authors response 4. We used simple and reproduceable reaction that could be scaled. However, we didn`t study this issue considering the aim of the study.

Q 5. The pharmacokinetic data presented are just predicted rather than experimentally validated, which weakens the strength of the claims regarding the compounds' drug­likeness.

Authors response 5. We agree with opinion of the reviewer. Present paper is preliminary study, thus we limited pharmacokinetics study by in silico methods.

Q 6. Write the methods section in the past tense to show what has been performed to get the results.

Authors response 6. We wrote methods section in the past tense.

Q 7. se standard units and symbols throughout the manuscript, such as mL instead of ml, ng/L instead of ng/1, nM/mL instead of nM/ml, etc.

Authors response 7. We made corrections recommended by the reviewer.

Q 8. Please also comment in the manuscript on the safety, effectiveness, side effects, and cost-effectiveness of the tested drugs compared to the available anti-inflammatory drugs in the market.

Authors response 8. This paper does not operate with data that allow to make conclusions about safety, effectiveness, side effects, and cost-effectiveness of the studied compounds.

Q 9. What are the possible challenges in bringing the tested compounds to the market as alternative anti-inflammatory drugs?

Authors response 9. This paper does not operate with data that allow to make conclusions about management of the product based on the studied compounds.

Round 2

Reviewer 1 Report

Comments and Suggestions for Authors

I believe that the revised version of the manuscript is acceptable for publication in Pharmaceuticals.

I am requesting that the authors make one final minor modification: namely replacing the term “mercapto” with “sulfanyl” in the description of Methods A and B in Scheme 1.

Author Response

We thank the reviewer for the feedback

Q1. I believe that the revised version of the manuscript is acceptable for publication in Pharmaceuticals.

I am requesting that the authors make one final minor modification: namely replacing the term “mercapto” with “sulfanyl” in the description of Methods A and B in Scheme 1.

A1. The requested change was now made